# Comparisons of Viral Etiology and Outcomes of Hepatocellular Carcinoma Undergoing Liver Resection between Taiwan and Vietnam

**DOI:** 10.3390/v14112571

**Published:** 2022-11-20

**Authors:** Song-Huy Nguyen-Dinh, Wei-Feng Li, Yueh-Wei Liu, Chih-Chi Wang, Yen-Hao Chen, Jing-Houng Wang, Chao-Hung Hung

**Affiliations:** 1Liver Tumor Department, Cho Ray Hospital, Ho Chi Minh City 70250, Vietnam; 2Liver Transplant Center, Department of Surgery, Kaohsiung Chang Gung Memorial Hospital and Chang Gung University College of Medicine, Kaohsiung 83300, Taiwan; 3Division of Hematology-Oncology, Department of Internal Medicine, Kaohsiung Chang Gung Memorial Hospital and Chang Gung University College of Medicine, Kaohsiung 83300, Taiwan; 4Division of Hepatogastroenterology, Department of Internal Medicine, Kaohsiung Chang Gung Memorial Hospital and Chang Gung University College of Medicine, Kaohsiung 83300, Taiwan

**Keywords:** hepatocellular carcinoma, hepatitis B virus, hepatitis C virus, recurrence, survival

## Abstract

Epidemiologic data have suggested that etiologic variations of hepatocellular carcinoma (HCC) exist in different geographic areas, and might be associated with different outcomes. We compared the viral etiology, clinicopathological characteristics and surgical outcomes between 706 Taiwanese and 1704 Vietnamese patients with HCC undergoing liver resection. Vietnamese patients had a significantly higher ratio of hepatitis B virus (HBV) (*p* < 0.001) and a lower ratio of hepatitis C virus (HCV) (*p* < 0.001) and non-B non-C than Taiwanese patients. Among patients with HBV or non-B non-C, the mean age was younger in Vietnam than in Taiwan (*p* < 0.001, *p* = 0.001, respectively). The HCC patients in Vietnam had significantly higher serum alpha-fetoprotein (AFP) levels (*p* < 0.001), larger tumors (*p* < 0.001), and a higher ratio of macrovascular invasion (*p* < 0.001) and extrahepatic metastasis (*p* < 0.001), compared to those in Taiwan. Patients treated in Vietnam had a higher tumor recurrent rate (*p* < 0.001), but no difference in overall survival was found between both groups. In subgroup analysis, the recurrent rate of HCC was the highest in patients with dual HBV/HCV, followed by HCV or HBV, and non-B non-C (*p* < 0.001). In conclusion, although the viral etiology and clinicopathological characteristics of HCC differed, postoperative overall survival was comparable between patients in Taiwan and Vietnam.

## 1. Introduction

Hepatocellular carcinoma (HCC) is the sixth most commonly occurring malignancy, and one of the largest contributors of cancer-related death in the world [1]. Despite recent advances in surgical technique and diagnostic modality, the overall five-year survival rate of HCC is still unsatisfactory [2]. In some developing countries of Asia and Africa, the annual incidence and mortality of HCC have been increasing [3]. Currently, chronic infection with hepatitis B virus (HBV) or hepatitis C virus (HCV) remains the most important risk factors for HCC [4]. Nevertheless, some other factors, include alcohol consumption, metabolic diseases, such as diabetes mellitus and obesity, and environmental toxins, such as alfatoxin, have become increasingly important etiologic factors [5,6,7].

Several epidemiologic studies have suggested that HCC in different geographic areas of the world may represent different forms of the disease [3]. It is worth investigating whether the regional differences in HCC characteristics and outcomes are due to different etiologies and underlying liver damage. HCC is highly prevalent in the Asia–Pacific region, where HBV infection plays the predominant role in the development of HCC [8]. In Vietnam, HCC is a serious public health issue, with age-adjusted incidence rates of over 20 per 100,000 people [3,9]. Moreover, the estimated incidence of HBV-related HCC is predicted to continue rising from 9400 in 1990 to 25,000 in 2025, even though universal infant HBV vaccination has been conducted [10,11]. In Taiwan, HCC is a leading cause of cancer-related mortality, and HBV infection is the most important viral etiology. As time moves on, the proportion of HCV-related HCC has been gradually increasing [12,13].

We, therefore, conducted a retrospective cohort study to compare the viral etiologies, clinicopathological characteristics and surgical outcomes, including overall survival (OS) and recurrence-free survival (RFS), after resection in patients with HCC treated at two tertiary centers in Taiwan and Vietnam.

## 2. Materials and Methods

Between January 2014 and December 2018, a total of 706 patients in Taiwan and 1704 patients in Vietnam who underwent attempted curative resection for HCC as initial treatment were enrolled. Kaohsiung Chang Gung Memorial Hospital is currently the largest one in southern Taiwan, at which the patients come from all areas of southern Taiwan [12]. Cho Ray hospital is the largest tertiary referral hospital in southern and central Vietnam, with 24,000 cases of HCC seen over a seven-year period [14]. Clinical and pathological data from all patients, including age, gender, viral hepatitis markers, preoperative serum alpha-fetoprotein (AFP) level, tumor size, tumor number, macrovascular invasion and extrahepatic metastasis were collected. This study was approved by the institutional review board of the hospital, and was conducted in accordance with the principles of the Declaration of Helsinki and the International Conference on Harmonization for Good Clinical Practice.

Data are presented as mean ± standard deviation (SD). Comparisons of differences in categorical data between groups were performed using the chi-square test. Distributions of continuous variables were analyzed by the *t*-test or one-way ANOVA test with least significant difference (LSD) post-hoc correction between groups where appropriate. The starting date of follow-up of each patient began at the time of liver resection. OS or RFS were measured until death from any cause or the last follow-up. Survival curves were constructed using Kaplan–Meier curves and the differences between groups were compared using the log-rank test. Furthermore, Cox proportional hazards models were used to compute the hazard ratios (HRs) with 95% confidence interval (CI), after adjustment for potential confounders. All of these analyses were carried out using statistical software (SPSS 15.0), and statistical significance was defined as a two-tailed *p* value of <0.05.

## 3. Results

### 3.1. Baseline Characteristics

The comparison of baseline characteristics of patients and tumors between Taiwan and Vietnam are shown in Table 1. There was a male predominance in both groups; whereas patients in Vietnam had a higher male to female ratio than those in Taiwan (*p* < 0.001). The HCC patients in Vietnam were younger (*p* < 0.001), and had significantly higher serum AFP level (*p* < 0.001), larger size of tumor (*p* < 0.001) and tended to have a single tumor (*p* < 0.001), having macrovascular invasion (*p* < 0.001) and extrahepatic metastasis (*p* < 0.001), compared to those in Taiwan. The most common viral etiology was HBV in both groups, but Vietnamese patients had a significantly higher ratio of HBV (*p* < 0.001) and a lower ratio of HCV (*p* < 0.001) and non-B non-C than Taiwanese patients.

### 3.2. Differences in Age and Gender According to Viral Etiology of HCC

Table 2 shows the differences in age and gender according to viral etiology of HCC between Taiwan and Vietnam. The mean age of HBV patients was significantly younger than that of patients with HCV (both *p* < 0.001) or non-B non-C (both *p* < 0.001) in both groups. Among patients with HBV or non-B non-C, the mean age was younger in Vietnam than in Taiwan (*p* < 0.001, *p* = 0.001, respectively), whereas there was no difference among those with HCV. The male-to-female ratio was higher in HBV than in HCV patients (*p* < 0.001) only in the Taiwanese group.

### 3.3. Factors Associated with RFS

As shown in Figure 1A, 332 patients in Taiwan and 1049 patients in Vietnam had tumor recurrence during the follow-up period (*p* < 0.001). In Table 3, multivariate Cox proportional hazards analysis showed the significant risk factors of tumor recurrence for HCC patients receiving hepatic resection were the following: site (HR: 2.06, 95% CI: 1.79–2.37; *p* < 0.001), male gender (HR: 1.37, 95% CI: 1.17–1.59; *p* < 0.001), higher AFP (≥400 ng/mL) (HR: 1.44, 95% CI: 1.29–1.61; *p* < 0.001), larger tumor size (≥5 cm) (HR: 1.37, 95% CI: 1.21–1.55; *p* < 0.001), multiple tumors (HR: 1.45, 95% CI: 1.27–1.67; *p* < 0.001), macrovascular invasion (HR: 2.17, 95% CI: 1.83–2.57; *p* < 0.001) and extrahepatic metastasis (HR: 2.20, 95% CI: 1.67–2.90; *p* < 0.001).

In regard to the viral etiology contributing to tumor recurrence, we, therefore, analyzed the patients who were classified into four groups accordingly. As shown in Figure 2, the incidence of HCC recurrence was the highest in patients with dual HBV/HCV infection, followed by HCV or HBV mono-infection, and non-B non-C (*p* < 0.001).

### 3.4. Factors Associated with OS

The OS between Taiwan and Vietnam was not different, as shown in Figure 1B. In Table 4, the significant factors associated with OS in HCC patients receiving hepatic resection were the following male gender (HR: 1.39, 95% CI: 1.08–1.81; *p* = 0.012), higher AFP (≥400 ng/mL) (HR: 1.60, 95% CI: 1.32–1.93; *p* < 0.001), larger tumor size (≥5 cm) (HR: 1.40, 95% CI: 1.14–1.72; *p* < 0.001), multiple tumors (HR: 1.79, 95% CI: 1.45–2.17; *p* < 0.001), macrovascular invasion (HR: 2.81, 95% CI: 2.18–3.62; *p* < 0.001) and extrahepatic metastasis (HR: 2.70, 95% CI: 1.82–4.01; *p* < 0.001), based on multivariate Cox proportional hazards analysis. The viral etiology was not associated with OS. 

## 4. Discussion

This present study compared the viral etiology, clinicopathological characteristics and surgical outcomes of HCC patients receiving liver resection between Taiwan and Vietnam in the Asia–Pacific region, where the prevalence of HCC is the highest worldwide. We demonstrated that HCC patients in Vietnam were younger, and had more aggressive tumor characteristics, such as higher serum AFP level, larger size of tumor, more macrovascular invasion and more extrahepatic metastasis, compared to those in Taiwan. The patients treated in Vietnam had a significant tumor recurrent rate after resection by multivariate analysis, in addition to other common risk factors, including male gender, higher AFP, larger tumor size, multiple tumors, macrovascular invasion and extrahepatic metastasis. However, despite these differences, the postoperative OS was compatible between Taiwan and Vietnam. Although the definite reason was unclear, this might be explained by the shorter mean follow-up period in Vietnam than in Taiwan. 

Globally, the male-to-female ratio of HCC patients has been reported to range from 2:1 to 5:1, with the variations associated with dominant hepatitis virus in the population and the existence of other risk factors [15]. In this study, the male-to-female ratio was higher in HBV than in HCV patients in the Taiwanese group, but not in the Vietnamese group. This finding could be partially explained by the fact that Vietnamese patients had a lower ratio of HCV infection than those in Taiwan. On the other hand, we found that HBV patients was significantly younger than patients with HCV or non-B non-C among both Taiwanese and Vietnamese patients. These data were compatible with those reported in previous studies [12,13], suggesting that age distribution of HCC depended on gender, etiology and region. However, in particular, the mean age was younger in Vietnam than in Taiwan among patients with HBV or non-B non-C, indicating that some risk factors, other than the hepatitis virus, such as agricultural use of organophosphorous pesticides or alfatoxin, should be considered in Vietnam [16,17].

Several clinicopathological factors have been constantly reported to have prognostic value following liver resection of HCC [12,18,19]. In this study, we demonstrated that male gender, higher AFP (≥400 ng/mL), larger tumor size (≥5 cm), multiple tumors, macrovascular invasion and extrahepatic metastasis were independent factors associated with poorer OS. Our data were consistent with those reported in most studies. In contrast, there was no significant difference in OS between Taiwan and Vietnam.

Whether different viral etiologies have an impact on long-term outcomes in patients undergoing liver resection for HCC is unclear. Cescon et al. reported that RFS was significantly higher in non-B non-C patients compared with those with HBV [20]. A recent study demonstrated that non-B non-C HCC occurs mostly in elderly patients, and OS rate is significantly higher than that of HBV–HCC patients [21]. In regard to dual HBV/HCV infection, previous meta-analyses have confirmed that patients with dual HBV/HCV infection have an increased risk of HCC, whereas data about dual HBV/HCV infection in association with surgical outcomes in HCC patients remain limited. Our data showed that the recurrent rate of HCC after liver resection was the highest in patients with dual HBV/HCV infection, followed by HCV or HBV mono-infection, and NBNC. This result was consistent with a recent report showing that the RFS rates in patients with HBV/HCV–HCC were significantly poorer than those with HBV–HCC in a propensity score matching cohort [22]. The reason why different etiologies impact on the HCC recurrence after liver resection remains unclear. A recent report indicated the key role of regulatory T cells in the HCC immune microenvironment and their presence has been correlated with tumor progression, and invasiveness, as well as metastasis [23]. However, our study did not confirm the association of viral etiology with OS. This issue should be further clarified due to limited cases of dual HBV/HCV infection in our study. 

The strength of this study was our large sample size of patients who were enrolled from two large tertiary referral centers. However, our study has the common limitations associated with its retrospective study design with some incomplete patient records and potential selection bias. In addition, our study did not analyze the treatment effect of antivirals such as nucleos(t)ide analogs for HBV, and interferon-based therapies or direct acting antivirals for HCV. Moreover, our data, such as the data on two-year recurrence and three-year survival, were compatible with those previously reported in a meta-analysis of single HCV-untreated studies, evaluating outcomes after curative treatments of HCV-related HCC [24,25]. Our study also did not analyze the effect of other treatment procedures for recurrent lesions on OS. Further studies are necessary to clarify these weak points. 

## 5. Conclusions

This study identified the differences in viral etiology and clinicopathological characteristics, as well as surgical outcomes, between Taiwan and Vietnam. Although HCC patients in Vietnam had more aggressive tumor characteristics and higher tumor recurrence rates after resection, there was no significance in postoperative OS between Taiwan and Vietnam.

## Figures and Tables

**Figure 1 viruses-14-02571-f001:**
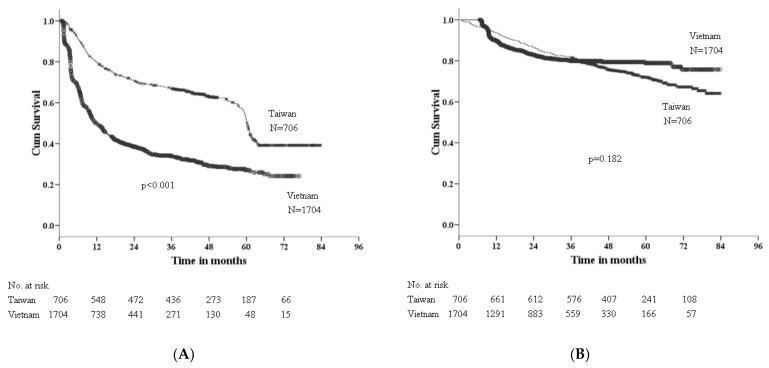
(**A**). Recurrence-free survival after resection of HCC, stratified by site of treatment. (**B**). Overall survival after resection of HCC, stratified by site of treatment.

**Figure 2 viruses-14-02571-f002:**
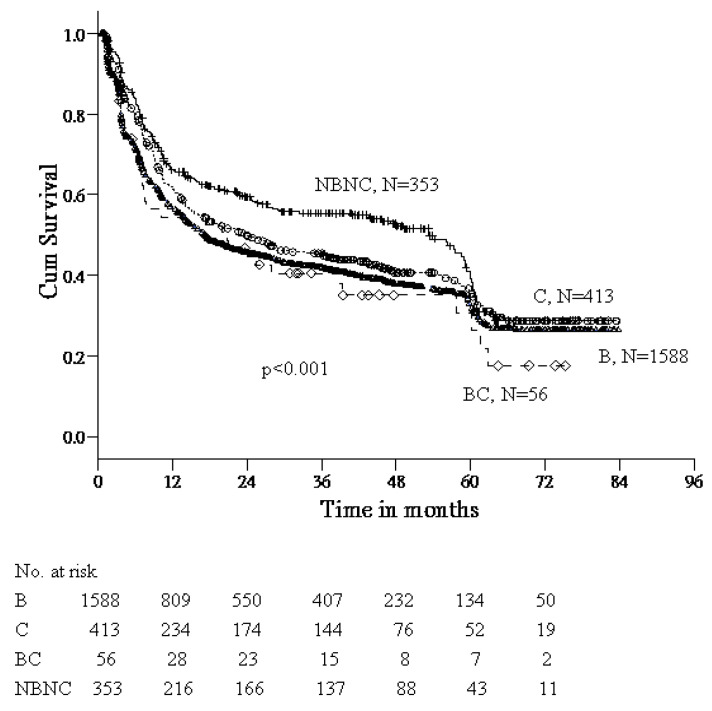
Recurrence-free survival after resection of HCC, stratified by viral etiology.

**Table 1 viruses-14-02571-t001:** Comparison of patient and tumor characteristics.

Variable	Taiwan		Vietnam		*p*-Value
	*n*	%	*n*	%	
Total	706		1704		
Gender					<0.001
Male	548	(77.6)	1433	(84.1)	
Female	158	(22.4)	271	(15.9)	
Age (years)					<0.001
<60	289	(40.9)	1064	(62.4)	
≥60	417	(59.1)	640	(37.6)	
Mean (SD)	60.9	(11.0)	54.6	(12.3)	
Viral hepatitis					<0.001
HBV	342	(48.4)	1246	(73.1)	
HCV	197	(27.9)	216	(12.7)	
HBV/HCV	20	(2.8)	36	(2.1)	
Non HBV/HCV	147	(20.8)	206	(12.1)	
Alpha-fetoprotein (ng/mL)					<0.001
<400	567	(80.3)	931	(54.6)	
≥400	138	(19.7)	773	(45.4)	
Mean (SD)	4653	(30,824)	13,093	(54,535)	
Tumor size (cm)					<0.001
<5	469	(66.4)	394	(23.1)	
≥5	237	(33.6)	1310	(76.9)	
Mean (SD)	4.9	(3.8)	7.2	(3.7)	
Tumor number					<0.001
Single	487	(72.7)	1457	(85.5)	
Multiple	183	(27.3)	247	(14.5)	
Macrovascular invasion					<0.001
No	644	(96.1)	1532	(90.0)	
Yes	26	(3.9)	172	(10.0)	
Extrahepatic metastasis					<0.001
No	663	(99.0)	1644	(96.5)	
Yes	7	(1.0)	60	(3.5)	

**Table 2 viruses-14-02571-t002:** Differences in age and gender according to viral etiology of HCC.

	HBV	HCV	B + C	NBNC	*p*-Value
Taiwan					
Case no.	342	197	20	147	
Age (years) *	57.5 ± 11.3 ^acf^	65.3 ± 8.3 ^ad^	59.5 ± 9.1 ^d^	63.1 ± 11.4 ^ch^	<0.001
Gender (M/F)	287/55 (5.2) ^a^	126/71 (1.8) ^aeg^	17/3 (5.7)	118/29 (4.1) ^e^	<0.001
Vietnam					
Case no.	1246	216	36	206	
Age (years) *	52.0 ± 11.8 ^acf^	65.2 ± 7.5 ^ad^	56.8 ± 13.1 ^d^	59.0 ± 11.8 ^ch^	<0.001
Gender (M/F)	1057/189 (5.6)	175/41 (4.3) ^g^	33/3 (11)	168/38 (4.4)	0.206

* Mean ± SD; *p*-value by one-way ANOVA test with least significant difference (LSD) post-hoc correction or *x*^2^ test. ^a^ Significant differences between HBV and HCV; ^b^ Significant differences HBV and B + C; ^c^ Significant differences between HBV and Non-B, Non-C; ^d^ Significant differences between HCV and B + C; ^e^ Significant differences HCV and Non-B, Non-C; ^f^ Significant differences between Taiwan and Vietnam HBV-HCC; ^g^ Significant differences between Taiwan and Vietnam HCV–HCC; ^h^ Significant differences between Taiwan and Vietnam NBNC–HCC.

**Table 3 viruses-14-02571-t003:** Risk of recurrence-free survival for HCC patients receiving hepatic resection.

Variable	HR (95% CI)	*p*-Value	aHR (95% CI) *	*p*-Value
Site		<0.001		<0.001
Taiwan	Reference		Reference	
Vietnam	2.44 (2.15–2.77)		2.06 (1.79–2.37)	
Sex		0.015		<0.001
Female	Reference		Reference	
Male	1.38 (1.20–1.60)		1.37 (1.17–1.59)	
Age (years)		0.003		
<60	Reference			
≥60	0.86 (0.77–0.95)			
HBsAg		<0.001		
Positive	1.26 (1.12–1.41)			
Negative	Reference			
Anti-HCV		0.509		
Positive	0.96 (0.84–1.09)			
Negative	Reference			
Alpha-fetoprotein (ng/mL)		<0.001		<0.001
<400	Reference		Reference	
≥400	1.82 (1.63–2.02)		1.44 (1.29–1.61)	
Tumor size (cm)		<0.001		<0.001
<5	Reference		Reference	
≥5	1.98 (1.76–2.22)		1.37 (1.21–1.55)	
Tumor number		<0.001		<0.001
Single	Reference		Reference	
Multiple	1.37 (1.20–1.56)		1.45 (1.27–1.67)	
Macrovascular invasion		<0.001		<0.001
No	Reference		Reference	
Yes	2.82 (2.38–3.33)		2.17 (1.83–2.57)	
Extrahepatic metastasis		<0.001		<0.001
No	Reference		Reference	
Yes	2.85 (2.17–3.75)		2.20 (1.67–2.90)	

* Multivariable analysis by Cox proportional hazards model.

**Table 4 viruses-14-02571-t004:** Risk for HCC patients receiving hepatic resection.

Variable	HR (95% CI)	*p*-Value	aHR (95% CI) *	*p*-Value
Site		0.182		
Taiwan	Reference			
Vietnam	0.88 (0.73–1.06)			
Sex		0.015		0.012
Female	Reference		Reference	
Male	1.37 (1.06–1.76)		1.39 (1.08–1.81)	
Age (years)		0.652		
<60	Reference			
≥60	1.04 (0.87–1.25)			
HBsAg		0.847		
Positive	0.98 (0.81–1.18)			
Negative	Reference			
Anti-HCV		0.711		
Positive	0.96 (0.77–1.20)			
Negative	Reference			
Alpha-fetoprotein (ng/mL)		<0.001		<0.001
<400	Reference		Reference	
≥400	1.87 (1.56–2.24)		1.60 (1.32–1.93)	
Tumor size (cm)		<0.001		0.001
<5	Reference		Reference	
≥5	1.72 (1.41–2.10)		1.40 (1.14–1.72)	
Tumor number		<0.001		<0.001
Single	Reference		Reference	
Multiple	2.08 (1.69–2.56)		1.79 (1.45–2.17)	
Macrovascular invasion		<0.001		<0.001
No	Reference		Reference	
Yes	3.79 (2.96–4.84)		2.81 (2.18–3.62)	
Extrahepatic metastasis		<0.001		<0.001
No	Reference		Reference	
Yes	3.62 (2.45–5.35)		2.70 (1.82–4.01)	

* Multivariable analysis by Cox proportional hazards model.

## Data Availability

The data presented in this study are available on reasonable request to the first author and corresponding author.

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
