# Peer review of "Comparisons of Viral Etiology and Outcomes of Hepatocellular Carcinoma Undergoing Liver Resection between Taiwan and Vietnam"

_viruses, 2022, doi:10.3390/v14112571_

Round 1

Reviewer 1 Report

In this manuscript, the authors showed the differences in viral etiology and clinicopathological characteristics as well as surgical outcomes between Taiwan and Vietnam. Interestingly, HCC patients in Vietnam had more aggressive tumor characteristics and higher tumor recurrence rate after resection; however, there was no significance in postoperative OS between Taiwan and Vietnam. This is an interesting study; however, the reason why the OS after surgical resection were comparable between the two countries are not discussed in this manuscript. The authors should discuss this point. In addition, the RFS of Taiwanese HCC patients suddenly decreased in around 60 months after surgery (Fig. 1A). The authors should explain the reason. Besides, the number of patients who underwent anti-viral therapies, including nucleoside analogues and direct acting antivirals, should be presented in Table 1.

Author Response

Point 1:

In this manuscript, the authors showed the differences in viral etiology and clinicopathological characteristics as well as surgical outcomes between Taiwan and Vietnam. Interestingly, HCC patients in Vietnam had more aggressive tumor characteristics and higher tumor recurrence rate after resection; however, there was no significance in postoperative OS between Taiwan and Vietnam. This is an interesting study; however, the reason why the OS after surgical resection were comparable between the two countries are not discussed in this manuscript. The authors should discuss this point.

Response 1:

Thank you very much for your kind suggestion. We have described the possible reason in page 8, line 12~13. "Although the definite reason was unclear, this might be explained by the shorter mean follow-up period in Vietnam than in Taiwan."

Point 2:

In addition, the RFS of Taiwanese HCC patients suddenly decreased in around 60 months after surgery (Fig. 1A). The authors should explain the reason.

Response 2:

Thank you very much for your valuable comment. The mean RFS in Taiwanese patients was 39 ± 24 months. Seventy-three percent (518/706) of patients had a RFS lower than 60 months. Thus, smaller case number would lead to suddenly decreased RFS after 60 months of follow-up.    

Point 3

Besides, the number of patients who underwent anti-viral therapies, including nucleoside analogues and direct acting antivirals, should be presented in Table 1.

Response 3:

We are sorry that data about antiviral therapies were not avaiable from the incomplete patient records in Vietnam. We have acknowledged this study limitation in page 9, line 16~18.

Reviewer 2 Report

In this study, the authors compared the viral etiology, clinicopathological characteristics, and surgical outcomes between 706 Taiwanese and 1704 Vietnamese patients with HCC undergoing liver resection. 

They found that Vietnamese patients had a significantly higher ratio of hepatitis B virus (HBV) (p<0.001) and a lower ratio of hepatitis C virus (HCV) (p<0.001) and non-B non-C than Taiwanese patients. Among patients with HBV or non-B non-C, the mean age was younger in Vietnam than in Taiwan (p<0.001, p=0.001, respectively). Interestingly, the HCC patients in Vietnam had significantly higher serum alpha-fetoprotein (AFP) level (p<0.001), larger tumor (p<0.001), and higher ratio of macrovascular invasion (p<0.001) and extrahepatic metastasis (p<0.001) compared to those in Taiwan.

Patients treated in Vietnam had a higher tumor recurrent rate (p<0.001) but no difference in overall survival was found. In subgroup analysis, the recurrent rate of HCC was the highest in patients with dual HBV/HCV, followed by HCV or HBV, and non-B non-C (p<0.001). They concluded that although the viral etiology and clinicopathological characteristics of HCC differed, postoperative overall survival was comparable between patients in Taiwan and Vietnam.

The study is of interest and provides novel findings. However, some points deserve further data and should be addressed.

-Study population: Non HBV/HCV patients. Please, specify etiology of this patients subgroup.

-The study did not analyze the treatment effect of antivirals such as nucleos(t)ide analogues for HBV, and interferon-based therapies, or direct acting antivirals for HCV. The authors should recall previous literature data on the effect of antiviral treatments on HCC risk and recurrence after curative treatments, as previously demonstrated (A meta-analysis of single HCV-untreated arm of studies evaluating outcomes after curative treatments of HCV-related hepatocellular carcinoma. Liver Int. 2017 Aug;37(8):1157-1166; The changing scenario of hepatocellular carcinoma in Italy: an update. Liver Int. 2021 Mar;41(3):585-597. ).

- The authors should discuss the impact of different etiology on the HCC risk since it has been reported that HCC risk is higher in viral (HBV and HCV) chronic liver diseases than in other non-viral liver diseases as autoimmune liver diseases, as recently reported (Hepatocellular carcinoma in viral and autoimmune liver diseases: Role of CD4+ CD25+ Foxp3+ regulatory T cells in the immune microenvironment. World J Gastroenterol. 2021 Jun 14;27(22):2994-3009. ). 

Author Response

Point 1:

Study population: Non HBV/HCV patients. Please, specify etiology of this patients subgroup.

Response 1:

We are sorry that data about the details of non HBV/HCV therapies were not avaiable from the incomplete patient records in Vietnam. We have acknowledged this study limitation in page 9, line 17-19.

Point 2:

The study did not analyze the treatment effect of antivirals such as nucleos(t)ide analogues for HBV, and interferon-based therapies, or direct acting antivirals for HCV. The authors should recall previous literature data on the effect of antiviral treatments on HCC risk and recurrence after curative treatments, as previously demonstrated (A meta-analysis of single HCV-untreated arm of studies evaluating outcomes after curative treatments of HCV-related hepatocellular carcinoma. Liver Int. 2017 Aug;37(8):1157-1166; The changing scenario of hepatocellular carcinoma in Italy: an update. Liver Int. 2021 Mar;41(3):585-597. ).

Response 2:

Thank you very much for your valuable comment. We have discussed the previous literature data on the effect of antiviral treatments on HCC recurrence after curative treatments in page 9, line 21-24 " Moreover, our data such as 2-year recurrence and 3-year survival were compatible with those reported previously in a meta-analysis of single HCV-untreated arm of studies evaluating outcomes after curative treatments of HCV-related HCC." Two new references 23,24 have been cited.

Point 3:

The authors should discuss the impact of different etiology on the HCC risk since it has been reported that HCC risk is higher in viral (HBV and HCV) chronic liver diseases than in other non-viral liver diseases as autoimmune liver diseases, as recently reported (Hepatocellular carcinoma in viral and autoimmune liver diseases: Role of CD4+ CD25+ Foxp3+ regulatory T cells in the immune microenvironment. World J Gastroenterol. 2021 Jun 14;27(22):2994-3009. ). 

Response 3:

Thank you very much for your valuable comment. We have discussed the impact of different etiology on the HCC recurrence after liver resection in page 9, line 10-13. " The reason why different etiologies have impact on the HCC recurrence after liver resection remains unclear. A recent report has shown the key role of regulatory T cells in HCC immune microenvironment and their presence has been correlated with tumor progression, invasiveness, as well as metastasis [23]." A new reference 23 has been cited.